# Meta-Quantitative Trait Loci Analysis and Candidate Gene Mining for Drought Tolerance-Associated Traits in Maize (*Zea mays* L.)

**DOI:** 10.3390/ijms25084295

**Published:** 2024-04-12

**Authors:** Ronglan Li, Yueli Wang, Dongdong Li, Yuhang Guo, Zhipeng Zhou, Mi Zhang, Yufeng Zhang, Tobias Würschum, Wenxin Liu

**Affiliations:** 1Key Laboratory of Crop Heterosis and Utilization, The Ministry of Education, Key Laboratory of Crop Genetic Improvement, Beijing Municipality, National Maize Improvement Center, College of Agronomy and Biotechnology, China Agricultural University, Beijing 100193, China; 2Sanya Institute of China Agricultural University, Sanya 572025, China; 3Institute of Plant Breeding, Seed Science and Population Genetics, University of Hohenheim, 70599 Stuttgart, Germany

**Keywords:** maize, drought tolerance, meta-QTL, candidate genes

## Abstract

Drought is one of the major abiotic stresses with a severe negative impact on maize production globally. Understanding the genetic architecture of drought tolerance in maize is a crucial step towards the breeding of drought-tolerant varieties and a targeted exploitation of genetic resources. In this study, 511 quantitative trait loci (QTL) related to grain yield components, flowering time, and plant morphology under drought conditions, as well as drought tolerance index were collected from 27 published studies and then projected on the IBM2 2008 Neighbors reference map for meta-analysis. In total, 83 meta-QTL (MQTL) associated with drought tolerance in maize were identified, of which 20 were determined as core MQTL. The average confidence interval of MQTL was strongly reduced compared to that of the previously published QTL. Nearly half of the MQTL were confirmed by co-localized marker-trait associations from genome-wide association studies. Based on the alignment of rice proteins related to drought tolerance, 63 orthologous genes were identified near the maize MQTL. Furthermore, 583 candidate genes were identified within the 20 core MQTL regions and maize–rice homologous genes. Based on KEGG analysis of candidate genes, plant hormone signaling pathways were found to be significantly enriched. The signaling pathways can have direct or indirect effects on drought tolerance and also interact with other pathways. In conclusion, this study provides novel insights into the genetic and molecular mechanisms of drought tolerance in maize towards a more targeted improvement of this important trait in breeding.

## 1. Introduction

Maize (*Zea mays* L.) is a globally important crop and serves as food, fodder, and industrial raw material [1,2,3]. Due to its broad adaptation, maize grows in different agro-ecological zones around the world, contributing approximately 21% of the global food production [4]. However, drought as the primary abiotic stress causes significant damage to maize production [5,6]. Therefore, the development of drought-tolerant maize varieties is pivotal to ensure stable maize yields for global food production.

Drought tolerance is a complex quantitative trait that is regulated by numerous genes with minor effects. It negatively impacts various agronomic traits of maize, including plant height, ear height, anthesis to silking interval, 1000 kernel weight, and grain yield. Water scarcity during development reduces plant and ear height, leading to insufficient photosynthesis [7]. Drought stress can also lead to sterile pollen production during flowering [8] as well as the increase in the anthesis-silking interval [9] and poor development of grain filling [10], which ultimately results in the decrease in yield. Previous studies have evaluated these traits under drought conditions to identify drought-tolerant germplasm for breeding [11].

Studies on drought tolerance of maize have also included quantitative trait locus (QTL) mapping. For example, Guo et al. [12] used a recombinant inbred line (RIL) population to perform QTL mapping on flowering time, plant height, grain yield, and yield component traits. Ana et al. [13] identified 43 grain yield and morphological trait QTL on all maize chromosomes except chromosome 9, and Zhao et al. [14] identified 69 QTL under drought stress and control conditions, which were associated with plant height, ear height, anthesis-silking interval, 100 grain weight, kernel weight, and kernel length. However, due to the low accuracy of most initially located QTL, only a few have been verified and subsequently applied in marker-assisted breeding in maize. Consequently, it is necessary to identify consistent QTL and to reduce their confidence interval in order to improve their utilization in breeding. This can be achieved by meta-QTL analysis, which is an approach to integrate QTL data from different studies, in order to identify meta-QTL with a reduced confidence interval [15]. This approach has the advantage that it allows the integration of the results of different mapping populations, different molecular markers, as well as different genetic linkage maps. Consequently, this method has been widely used in plant genetics and breeding, for example for drought tolerance in foxtail millet [16], yield traits under drought in rice [17], and yield-related traits of wheat [18]. In maize, meta-QTL analysis has been successfully applied to traits associated with grain quality and yield [19], disease resistance [20,21], and nutrient utilization [22].

A better understanding of the genetic and molecular mechanisms of drought tolerance in maize is crucial for future breeding and a more targeted exploitation of genetic resources. In this study, we therefore performed meta-QTL analysis for drought tolerance in maize. In particular, our objectives were as follows: (1) to identify consensus genomic regions linked to drought tolerance through meta-analysis, (2) to support the identified meta-QTL by comparison with results from genome-wide association studies (GWAS), (3) to identify candidate genes within the MQTL regions by searching for rice homologous genes and by exploring the important MQTL regions, and (4) to further characterize the candidate genes of MQTL.

## 2. Results

### 2.1. Distribution of QTL for Drought Tolerance in the Maize Genome

For this meta-QTL study, 56 original QTL mapping studies on maize drought tolerance published between 1996 and 2023 were collected. By filtering according to the water treatments and by QTL quality information, 27 studies [9,12,13,23,24,25,26,27,28,29,30,31,32,33,34,35,36,37,38,39,40,41,42,43,44,45,46] with 511 QTL were selected for the meta-analysis (Appendix A). The population size of these studies ranged from 49 to 450, and the marker types were mainly SSR, AFLP, RFLP, and SNP markers. The number of QTL obtained from each study ranged from 11 to 92. Regarding the LOD scores of these QTL, the number of QTL with LOD less than 2 was 31, most of the QTL (329) had a LOD score between 2 and 4, and 119 QTL were between 4 and 6. The number of QTL with an LOD < 2 (31) or > 6 (32) were relatively low (Figure 1a). Regarding another key feature, the proportion of variance explained (PVE) by the QTL, most of the QTL (243) explained 5–10%, while the remainder of the three levels <5% (101), 10–15% (95), and >15% (72) appeared to be equivalent (Figure 1b). There were 24 traits measured under drought conditions, which can be divided into four categories (Appendix A). Grain yield component traits (GY) accounted for the highest proportion (59.9%), followed by flowering time-related traits (FR) (23.9%), and plant morphology traits (PM) (11.7%), while drought tolerance index (DTI) accounted for 4.5% (Figure 1c; Appendix A).

### 2.2. Meta-Analysis of QTL for Drought Tolerance in Maize

Meta-analysis of the 511 QTL associated with drought tolerance in maize identified a total of 83 MQTL (Figure 2a and Appendix A; Appendix A). The maximum and minimum number of MQTL per chromosome was 10 (Chr. 2 and 8) and 6 (Chr. 5), respectively. Regarding the number of initial QTL, Chr. 1 harbored the most with 96, which formed 9 MQTL after meta-analysis (Figure 2b). Except for MQTL4_2 and MQTL4_7, which were both formed by only a single initial QTL, all other MQTL were formed by at least two initial QTL. The two MQTL with the highest number of initial QTL are MQTL8_2 (located on Chr. 8 incorporating 21 initial QTL related to 11 traits) and MQTL5_3 (located on Chr. 5 that had 20 initial QTL from 12 traits). There were four MQTL with a very narrow confidence interval of less than 1 cM, MQTL2_5, MQTL3_7, MQTL4_9, and MQTL8_10, whose confidence interval was 0.6, 0.8, 0.9, and 0.4 cM, respectively, and which influenced 6, 11, 10, and 3 traits, respectively (Appendix A). The average confidence interval of MQTL were substantially decreased compared with the confidence interval of the initial QTL on every chromosome, ranging from a 2.5-fold reduction (Chr. 9) to a 6.5-fold reduction (Chr. 5), with an average value of 4.3-fold reduction across the whole genome (Figure 2c).

### 2.3. Confirmation of MQTL with GWAS Results

In order to provide additional confirmation of the identified meta-QTL, their physical positions were compared with the physical positions of marker–trait associations (MTAs) identified by GWAS. A total of 40 MQTL (upstream and downstream 500 kb regions) were found to overlap with 247 MTAs, which were identified in five GWAS on drought tolerance in maize [47,48,49,50,51] (Appendix A). Among them, MQTL1_3 overlapped with 68 MTAs and MQTL4_9 with 29 MTAs, followed by MQTL2_7, MQTL8_9, and MQTL2_10 overlapping with 18, 17, and 12 MTAs, respectively. Furthermore, there were 11 MQTL overlapping with 3–10 MTAs and 7 MQTL overlapping with 2 MTAs, while 17 MQTL only overlapped with 1 MTA. It is worth noting that the latter MTA-MQTL included five MQTL with very small confidence interval of less than 1 Mb (Figure 3).

### 2.4. Exploring Candidate Genes for Drought Tolerance in the MQTL Regions

Several genes related to drought tolerance in maize were found in the MQTL regions (Figure 2a; Appendix A). This includes *ZmWRKY79* in MQTL7_4 that impacts lateral roots, lower stomatal aperture, and water loss under drought stress [52], *ZmXerico2* of MQTL7_5 confers ABA hypersensitivity and improves water use efficiency by overexpression [53], *ZmTCP42* in MQTL7_7 is teosinte-branched 1/cycloidea/proliferating (TCP) plant-specific transcription factors and plays a positive role in drought tolerance [54], and *ZmPTF1* in MQTL9_1 is known to contribute to root development and to improve drought tolerance [55]. Furthermore, MQTL9_3 contains *ZmTIP1* that contributes to root hair elongation [56] and *ZmVPP1* enhances photosynthetic efficiency and root development [49], *ZmRtn16 of MQTL9_5*, encoding a reticulon-like protein, was found to contribute to drought resistance by facilitating the vacuole H+-ATPase activity [57], *ZmBSK1 of* MQTL9_9 positively affects drought tolerance in maize [58], and *ZmASR1,* that affects the synthesis of branched-chain amino acids and maintains maize grain yield under drought conditions, is located in MQTL10_2 [59]. In addition, some drought tolerance genes were searched and found along 500 kb upstream and downstream of the MQTL regions, such as *ZmLBD2* (MQTL1_1) [60], *ZmCIPK3* (MQTL1_3) [50], *ZmNF-YA3* (MQTL1_5) [61], *ZmEREBP60* (MQTL1_7) [62], *ZmNAC080308* (MQTL3_6) [63], *ZmPP2C-A* (MQTL4_9) [64], *ZmALDH22A1* (MQTL7_8) [65], and *ZmWRKY106* (MQTL8_10) [66].

In order to identify further candidate genes potentially related to drought tolerance traits in the MQTL regions, three methods were applied. First, maize genes homologous to 271 rice drought tolerance genes were searched. Based on the homology between maize and rice through protein alignment, a total of 63 orthologous maize genes were found in the MQTL regions (Appendix A). The 63 candidate genes had effects on similar drought tolerance-related traits in maize and rice. For example, *Zm00001d029740* (MQTL1_4) and *Zm00001d028999* (MQTL1_3) affect yield-related traits similar to their rice homologous genes *OsSCE1* and *OsNAC10*, respectively. *Zm00001d023420* (MQTL10_1) and its rice homologous gene *RCN1* both affect seed weight and thickness under drought stress. The gene *Zm00001d052537* located in the MQTL4_6 region and *OsCEN2* both affect flowering-related traits during drought stress (Appendix A). These results indicate that the functions of these candidate genes are likely conserved in maize and rice. As a second approach, we explored the genomic regions of breeders’ MQTL for candidate genes. MQTL with a confidence interval physical distance less than 1 Mb, a genetic map distance less than 4 cM and initial QTL number greater than 2, are called breeders’ MQTL [67]. There were eight breeders’ MQTL, namely MQTL4_9, MQTL5_6, MQTL7_1, MQTL8_2, MQTL8_6, MQTL8_7, MQTL8_10, and MQTL9_8 that were explored further. Based on gene annotation, 106 promising candidate genes were found for those MQTL. For the third approach, we explored candidate genes within the most promising MQTL with MTA hits. The MQTL that were matched with more MTAs have a higher probability of harboring functional genes related to drought tolerance. We set the threshold that a MQTL must have greater than three MTAs to be considered for this analysis, which left 14 MTA-MQTL, including MQTL1_3, MQTL2_3, MQTL2_5, MQTL2_7, MQTL2_10, MQTL3_4, MQTL4_9, MQTL6_7, MQTL7_1, MQTL7_6, MQTL8_8, MQTL8_9, MQTL9_9, and MQTL10_8. Except for MQTL4_9 and MQTL7_1, which coincided with the breeders’ MQTL, the remaining 12 MQTL with larger confidence intervals (> 4 cM) were used and 453 candidate genes were found in the 500 kb interval surrounding the MQTL peak. Given the two overlapping MQTL, the eight breeders’ MQTL and the 14 MTA-MQTL gave 20 MQTL that are most promising for drought tolerance and that are defined here as core MQTL (Figure 2a). These 20 core MQTL and the maize–rice orthologous genes were investigated and a total of 583 candidate genes were identified for them (Appendix A).

### 2.5. Functional Annotation of Candidate Genes

We next performed GO and KEGG pathway enrichment analysis of the identified 583 candidate genes to determine their functional classification. Among them, 360 genes with GO annotation were mainly related to biological processes (19 items), molecular functions (17 items), and cellular components (2 items) (Figure 4). The most abundant GO terms related to biological processes were cellular process (GO: 0009987, 236/360, 65.6%) and metabolic process (GO: 0008152, 203/360, 56.4%), biological regulation (GO: 0065007, 86/360, 23.9%), regulation of biological process (GO:0050789, 79/360, 21.9%), and response to stimulus (GO: 0050896, 66/360, 18.3%). GO terms related to molecular function, binding (GO: 0005488, 198/360, 55.0%), and catalytic activity (GO: 0003824, 162/360, 45.0%) were also highly enriched. Cellular anatomical entity (GO: 0110165, 273/360, 75.8%) and protein-containing complex (GO: 0032991, 43/360, 11.9%) were enriched in the cellular component’s annotation. The KEGG metabolic pathway is significantly enriched with signal transduction pathways and protein kinases, such as plant hormone signal transduction, mitogen-activated protein kinases (MAPK) signaling pathway-plant, and plant–pathogen interaction (Figure 5).

### 2.6. Expression Analysis of Candidate Genes

The expression characteristics of the identified candidate genes in major stages and tissues were analyzed through the public database qTeller. The results showed that 504 of the 583 candidate genes had expression levels, 398 genes had expression levels > 2 transcript per million (TPM), and 307 genes even had > 10 TPM in at least one tissue (Appendix A). Here, we focused on 169 candidate genes (63 maize–rice homologous genes and 106 candidate genes within the breeders’ MQTL regions), of which 104 candidate genes with highly specific expression (TPM > 2) in various tissues were visualized (Figure 6; Appendix A). According to their different expression patterns, these 104 candidate genes were divided into four categories (Figure 6). In the first category, the expression levels were high in almost all tissues and stages. Among them, *Zm00001d012675* (gst1-glutathione-S-transferase1) had the highest expression in all tissues and is involved in the regulation of diverse stress tolerances. The expression levels of *Zm00001d038709*, *Zm00001d009594*, *Zm00001d038543*, *Zm00001d048102*, and *Zm00001d052416* were also quite high in almost all tissues at each stage. The second type was highly expressed only in some tissues. For example, *Zm00001d018030*, which may be related to photosynthesis, has the highest expression in leaf tissues. *Zm00001d018744* and *Zm00001d042541* are highly expressed in tap roots, crown roots, and brace roots, and may affect root length and root diameter. The third type of genes was expressed in all tissues but the expression level was rather low, and the fourth type was expressed only in one or some tissues/stages but at a medium level.

## 3. Discussion

### 3.1. Characteristics of QTL and MQTL for Drought Tolerance in Maize

In the last three decades, a large number of QTL mapping studies have been carried out. Due to the different genetic material as well as the different types of molecular markers used in each study, the QTL results are often not comparable or transferable. Moreover, even for the same marker type, the genetic linkage maps are different, complicating comparisons of QTL positions, and the confidence intervals of most QTL are quite large, making the use of identified QTL in marker-assisted selection (MAS) inaccurate. Meta-QTL analysis can overcome those limitations and can combine QTL results from different environments and genetic backgrounds to locate consistent MQTL with high reliability. For example, Guo et al. [67] performed meta-analysis for chlorophyll traits in wheat with 411 original QTL and identified 56 consensus MQTL with an average confidence interval 3.2 times narrower than that of the original QTL. Sharma et al. [68] collected 523 QTL to carry out meta-analysis for silage quality traits in maize and also achieved substantial reductions in the size of the confidence intervals. Sethi et al. [19] conducted meta-analysis for grain quality and yield-related traits with 2974 initial QTL in maize and obtained a total of 68 MQTL with a mean physical confidence interval of 3.30 Mb. Concerning drought tolerance, Loni et al. [16] employed meta-analysis in foxtail millet with 448 initial QTL and identified 41 MQTL. In maize, Liu et al. [69] also conducted a meta-analysis for drought tolerance with 457 initial QTL, and 74 MQTL were found.

In our study, 511 initial QTL for grain yield and its component traits, flowering time-related traits, plant morphology traits, and drought tolerance index under drought conditions were collected for meta-analysis based on the public genetic map of IBM2 2008 Neighbors [70]. In total, 83 MQTL were identified with an average confidence interval of 9.3 cM, which is an on average 4.3-fold reduction compared to that of the original QTL. The average confidence interval appears slightly larger than that of the similar study on drought-tolerance QTL in maize, which is due to the employed reference genetic map on which the consensus QTL are projected. For our study, we used the IBM2 2008 Neighbors map that integrates different segregating generations but always with B73 and Mo17 as parents (https://maizegdb.org/data_center/reference?id=1204261 accessed on 1 October 2023) and has an average genetic map length of 789.8 cM per chromosome. Since this reference map has the same parents (B73 and Mo17) and integrates over 19,000 public molecular markers, we believe that it is more useful and reliable than other consensus maps that are built with completely different parental combinations.

Compared with the previous study [69], we identified 26 overlapping MQTL (Appendix A), and most of them have a shortened confidence interval. Furthermore, in addition to the collection of the three major types of traits GY, FR, and PM under drought conditions, we paid more attention to the collection of DTI which we believe can better reflect the drought tolerance of maize. Specifically, we identified 8 breeders’ MQTL with a narrower physical interval (< 1Mb) and 95% confidence interval < 4 cM. These MQTL are promising candidates for future gene cloning as well as being reliable for the marker-assisted breeding of drought-tolerant maize varieties.

### 3.2. Many MQTL Can Be Substantiated by GWAS Results

It is valuable to validate MQTL with MTAs identified in GWAS, as this provides further evidence for their stability and reliability for the candidate genes in these genomic regions. Sharma et al. [68] found a total of 51 MTAs co-localized with the 20 MQTL. Li et al. [71] validated 31 of the 64 MQTL with at least one MTA. In our study, we found that 247 published MTAs were co-localized with 40 MQTL (500 kb upstream and downstream of the MQTL) and thus, nearly half (48.19%, 40/83) of the MQTL were verified by results from GWAS. In particular, these MQTL are promising for a further characterization, up to the molecular cloning of the underlying genes towards a better understanding of the molecular processes of drought tolerance in maize.

### 3.3. The Role of Plant Hormone Signaling Pathways in Drought Tolerance in Maize

A total of 583 candidate genes for the MQTL were identified based on three methods. Their further characterization by KEGG revealed that the most enriched metabolic pathway was plant hormone signal transduction (Figure 5). Plant hormones include abscisic acid (ABA), auxin (IAA), brassinosteroid (BR), cytokinin (CTK), ethylene (ETH), gibberellin (GA), jasmonate (JA), salicylic acid (SA), and strigolactone (SL), which regulate diverse processes including plant growth and response to abiotic stress [72]. In our study, 19 candidate genes from 13 MQTL are involved in plant hormone signaling pathways, including ABA, IAA, BR, CTK, ETH, and SA (Table 1; Figure 7).

First, there are six candidate genes involved in the ABA pathway (Table 1; Figure 7a). ABA is mainly able to induce stomatal closure and reduce leaf expansion under stress, in addition to its role in signal transduction in plant tissues [73]. Two candidate genes in MQTL5_3, *Zm00001d016294* (*ZmPYL3*) and *Zm00001d016105* (*ZmPYL10*) may play an important role in drought tolerance in maize, as their overexpression can enhance ABA signal transduction, proline, and other drought-related genes [74]. Moreover, *ZmPYL3* and *ZmPYL10* are homologous to the rice drought-related genes, *OsPYL3* and *OsPYL9* (Appendix A). For MQTL8_3, *Zm00001d009747* was identified, which is member of the protein phosphatase 2C (PP2C) family and homologous to the rice drought tolerance gene *OsABIL2*. The overexpression of *OsABIL2* was shown to significantly change the stomatal density and root structure, causing a hypersensitivity to drought stress [75]. Wei et al. [76] used microarray and RNA-seq data to analyze the expression profiles of *ZmPPs* at different developmental stages in maize. For drought stress conditions, 13 genes were found to be differentially expressed in the leaf, out of which 10 were up-regulated. For MQTL5_1, *Zm00001d013201* was identified, which belongs to the SnRK2 kinase group. It is a key component of the ABA pathway and is regulated by ABA receptors (PYR/PYL) and by the PP2C (Figure 7b). Studies by Hasan et al. [77] have shown that when plants are subjected to drought stress, they produce more ABA, which leads to defensive stress responses and activates many SnRK2 through ABA-dependent or ABA-independent pathways. *Zm00001d013201* is homologous to the rice gene *OsSAPK8* (Appendix A), which can be strongly induced by abiotic stress, including low temperature, high salt stress, and drought [78]. For MQTL8_5, a member of the ABF transcription factors was found (*Zm00001d010638*), which is homologous to the rice drought tolerance gene *OsbZIP62*. *OsbZIP62* can interact with ABA-activated protein kinases 1, 2, 4, and 6, that phosphorylate *OsbZIP* in rice. Plants overexpressing *OsbZIP62* showed increased drought tolerance and a high salt stress tolerance [79]. The ABA signaling cascade then regulates the drought and osmotic stress response through the downstream MAPK signaling pathway (Figure 7b).

In addition, there are seven candidate genes involved in the IAA, BR, CTK, and ETH signaling pathways. For the IAA pathway, three candidate genes were identified, namely *Zm00001d007395* (MQTL2_10), *Zm00001d053815* (MQTL4_9), and *Zm00001d010697* (MQTL8_5). *Zm00001d007395* and *Zm00001d010697* are homologous to the rice drought tolerance gene *OsGH3.13*, that encodes an indole-3-acetic acid (IAA)-amide synthase. This gene is significantly induced under drought stress and enhances the drought tolerance of rice [80]. Interestingly, *Zm00001d007395* (*ZmGH3.13*) was found to be differentially expressed in maize seedlings under heat stress [81]. Feng et al. [82] identified 13 *ZmGH3* genes and based on gene structure and tissue-specific expression patterns concluded that *ZmGH3s* are involved in the tolerance of maize to abiotic stress. For MQTL9_9, *Zm00001d048345* (*ZmBsk1*) related to BR signaling was found that interacts with calcium/calmodulin (Ca^2+^/CaM)-dependent protein kinase (*ZmCCaMK*) and phosphorylates *ZmCCaMK*. Drought stress enhances the phosphorylation of *ZmCCaMK* by *ZmBSK1*, which has a positive effect on drought tolerance in maize [58]. Furthermore, *Zm00001d012005* found for MQTL8_8 is a histidine receptor kinase of the CTK pathway, which senses cytokinin signaling and promotes the autophosphorylation of histidine. For the ETH pathway, two genes were identified, namely *Zm00001d028974* for MQTL1_3 and *Zm00001d047563* for MQTL9_5. Their homologous rice drought tolerance gene is *OsEIL2*, which confers abiotic stress sensitivity by regulating *OsBURP16* [83]. Xu et al. [84] identified *Zm00001d028974* (*ZmEIL3)* as a candidate gene related to iron deficiency tolerance.

Last, there are six genes from the SA pathway. *Zm00001d012553* of MQTL8_9 is the octopine synthase binding factor 4, which was found to be up-regulated in maize kernels under different N rates [85]. The remaining five genes are pathogenesis-related proteins (PRP or PRs), namely *Zm00001d007448* (MQTL2_10), *Zm00001d018734*, *Zm00001d018737*, and *Zm00001d018738* (MQTL7_1), and *Zm00001d019364* (MQTL7_4). The rice drought tolerance homologous gene of *Zm00001d018734*, *Zm00001d018738*, and *Zm00001d019364* is *OsPR1a*, whose transcripts were found to be induced by abiotic stress treatments, such as water-deficient oxidative stress, indicating that PR proteins play an important role in abiotic stress adaptation in addition to plant defense responses to pathogens. Compared with wild-type plants, overexpression of *OsPR1a* in Arabidopsis could enhance the tolerance to salt and water stress [86]. The accumulation of PR proteins can be stimulated by pathogen infection, but also by abiotic stress [87], indicating that the production and accumulation of this protein plays an important role in resistance to both biotic and abiotic stress in plants.

### 3.4. Characterization of MQTL Candidate Genes and Their Roles in Maize Drought Tolerance

The maize drought tolerance genes *ZmWRKY79* [52], *ZmXerico2* [53], *ZmTCP42* [54], *ZmPTF1* [55], *ZmTIP1* [56], *ZmVPP1* [49], *ZmRtn16* [57], *ZmBSK1* [58], and *ZmASR1* [59] have been identified for the MQTL. Among them, *ZmASR1* affects the synthesis of branched-chain amino acids and maintains the grain yield of maize under drought conditions. *ZmASR1* is located in MQTL10_2, which is not only related to ear length and diameter, but also to flowering traits. This indicates that *ZmASR1* may regulate drought tolerance by affecting the flowering process and through this ultimately the grain yield of maize. Both *ZmPTF1* and *ZmTIP1* contribute to the development of roots, and the corresponding MQTL affect traits related to yield, flowering period, and seed setting rate, which may be related to the enhancement of roots and of drought tolerance, thus ensuring a normal flowering period and yield. The MQTL9_3 contains two related drought tolerance genes, *ZmTIP1* and *ZmVPP1*, involved in 16 initial QTL for a large range of ten different traits, indicating its complex regulatory role.

In this study, we also exploited the close evolutionary relationship between the genomes of the gramineous plants maize and rice, as the analysis of maize–rice homologous relationships can broaden our understanding of maize genes. For example, *OsSCE1* [88], *OsNAC10* [89], and *RCN1* [90] have been shown to affect drought tolerance in rice and have similar functions in maize, indicating that it is possible to identify candidate genes based on interspecific homology analysis. A total of 63 maize–rice orthologous drought tolerance genes were found in the MQTL genomic regions, which are relatively conserved and may therefore affect similar traits in maize (Appendix A).

By analyzing tissue-specific expression, we found that 398 candidate genes are highly specifically expressed in various tissues (TPM > 2) in leaves, roots, and grains (Appendix A), and 104 of them were visualized (Figure 6; Appendix A). These candidate genes have strong expression in tissues that may affect the drought tolerance of maize. For example, *Zm00001d012675* (glutathione-S-transferase1, GST1) has a strong expression in various tissues of maize. GST plays an important role in the defense system of organisms, and previous studies have shown that the overexpression of GST genes in Arabidopsis, rice, and wheat, led to an enhanced drought or salt tolerance [91,92,93]. The most strongly expressed gene in leaf tissue was *Zm00001d018030* (NDH subunit F6, NDF6). Zhang et al. [94] created maize plants lacking NDH function and observed a significant decrease in its growth, photosynthetic activity, and key photosynthetic protein levels. This indicates that the gene may affect photosynthesis and thereby crop drought tolerance. In summary, based on the functional exploration of maize–rice homologies and the analysis of tissue expression patterns, several high-confidence candidate genes for drought tolerance in maize were found in the MQTL regions.

### 3.5. The Homology among Plant Species Shows Promising Prospect to Gene Resource Mining

From genetics to breeding, gene resource mining is the first step. The second step is to assess the polymorphism among the natural population and verify the function of each gene. If there are many genes related to the same function, the third step is to diagram the temporal–spatial expression of those genes, and evaluate the synergism or antagonism among those genes. Finally, we can introduce those functional genes to breed ideal target variety. The first step gene resource mining is the basis and crucial to the study system.

Due to convergence in selection and domestication, there is a great deal of homology among crop species. A star gene, *KNR2*, shows convergent selection and orthologs between rice and maize [95]. Rice is a model plant in crops, and a large number of drought-tolerance genes were explored and verified, we found over 271 drought-tolerance genes from the China Rice Data Center (https://ricedata.cn/ontology/ontology.aspx?ta=TO:0000276 accessed on 1 January 2024). Basing on the homology between maize and rice, we identified 63 drought tolerance genes within the MQTL regions in maize. Apart from rice, the proximate species sorghum [96,97,98], and even some Gramineae grasses such as *Zea mays* ssp. *Mexicana* [99], could provide anti-stress genes for maize. It provides a significant prospect to explore new genes based on the homology among plant species.

## 4. Materials and Methods

### 4.1. Collection of QTL Information

In this study, QTL mapping studies on drought tolerance in maize were collected from the website of the China Knowledge Network (https://www.cnki.net/ accessed on 1 October 2023), PubMed (https://pubmed.ncbi.nlm.nih.gov/ accessed on 1 October 2023), and the Web of Science (https://www.webofscience.com/ accessed on 1 October 2023), using the keywords ‘maize’, ‘drought’, and ‘QTL’ for searching. The main QTL information of drought tolerance-related traits, including QTL name, trait, chromosome, position, LOD value, proportion of variance explained (PVE), confidence interval (CI), population type, mapping population size, and genetic map were collected. The phenotypes collected in the drought tolerance studies included grain yield (GY), 1000 kernel weight (KWT), kernel weight per ear (KWE), number of rows per ear (NRE), number of kernels per row (NKR), ear number (EN), kernel number (KN), ear length (EL), ear weight (EW), ear diameter (ED), kernel length (KL), kernel width (KW), kernel thickness (KT), cob weight (CW), cob diameter (CD), ear setting percentage (ES), anthesis to silking interval (ASI), male flowering (MF), flowering days (FD), female flowering (FF), tassel branch number (TBN), plant height (PH), ear height (EH), and drought tolerance index (DTI). The 24 phenotypic traits (Appendix A) were categorized into grain yield component traits (GY), flowering time-related traits (FR), plant morphology traits (PM), and drought tolerance index (DTI). IBM2 2008 Neighbors was downloaded from the website MaizeGDB (https://maizegdb.org/data_center/map?id=1140201 accessed on 1 October 2023) as a unified genetic map and reference map. The collected QTL were subjected to a quality check and QTL with a confidence interval > 200 cM, PVE < 1%, or LOD < 1.5 were removed. Moreover, only the phenotypic data under drought conditions were retained, while results from conditions with sufficient water were excluded.

### 4.2. Integration of QTL Information

According to the information requirements of QTL collected by Biomercator V4.2 software [100], the CI and PVE of QTL are two key parameters. The meta-analysis of QTL is mainly achieved through the QTL LOD score, PVE, position, and CI. If the collected QTL data lack the 95% CI, it is inferred according to the formula offered by Darvasi and Soller [101], where N is the size of the original mapping population:(1)CI=530/(N×PVE)
(2)CI=163/(N×PVE)Formula (1) is suitable for backcross and F_2_ mapping populations, and Formula (2) is suitable for RIL mapping populations.

### 4.3. Projection and Meta-Analysis of Initial QTL

The information of the complete genetic map and collected QTL was uploaded by genetic data loading, and the QTL were mapped to the reference map. The genetic linkage map of IBM2 Neighbors is a combination of the high-density molecular marker linkage map of maize (Intermated B73 × Mo17 Map; IBM) and other molecular marker linkage maps. The map contains 19,111 loci, including RFLP, SSR, and RAPD markers, gene and sequence probes, with a total length of 7898.35 cM (https://maizegdb.org/data_center/map accessed on 1 October 2023, last updated on August 10, 2022 by Marty Sachs). The number of meta-QTL on each chromosome was determined by five optional criteria: AIC (Akaike information criterion), AICc (AIC correction), AIC3 (AIC 3 candidate model), BIC (Bayesian information criterion), and AWE (average weight of evidence). Each model provides the most likely position and CI on the chromosome by Gaussian theorem, according to the maximum likelihood function ratio method. The best MQTL model was determined using the lowest value of the five optional criteria, which helped us to ascertain the number of generated MQTL [68]. Finally, the obtained CI of MQTL corresponded to the left and right markers according to IBM2 2008 Neighbors, and the physical position of the markers in the RefGen_v4 version was obtained in MaizeGDB (https://chinese.maizegdb.org/ accessed on 1 October 2023). If there was no physical position of the marker, the relevant physical position can be obtained through the primer sequence based on local BLASTN.

### 4.4. Verification of MQTL with GWAS Studies

In order to substantiate the identified MQTL, five independent maize genome-wide association studies (GWAS) on drought-tolerance were collected [47,48,49,50,51]. The physical position of significant marker–trait associations (MTAs) from these studies was compared with the physical positions of the MQTL. The overlapping MTAs or adjacent MTAs (within 500kb upstream and downstream of the MQTL regions) were considered as verification of the MQTL [19].

### 4.5. Mining and Functional Annotation of Candidate Genes

Three methods were used to explore candidate genes in the MQTL genomic regions: 

(1) The maize–rice homology was exploited and maize genes orthologous to rice drought tolerance-related genes were identified by sequence alignment. For this, we downloaded all published rice drought tolerance-related genes with functional verification from the China Rice Data Center (https://www.ricedata.cn/ accessed on 1 October 2023) and from various databases of the literature, and extracted the protein sequence of the rice drought tolerance genes through the Rice Genome Annotation Project (http://rice.uga.edu/analyses_search_blast.shtml accessed on 1 October 2023). The BLASTP alignment of the maize protein sequence was performed by inputting the rice protein sequence through the Phytozome website (https://phytozome-next.jgi.doe.gov/ accessed on 1 October 2023). The alignment criterion was Evalue < $1\times e^{-10}$ and identity > 40%.

(2) As a second approach, we explored candidate genes within the breeders’ MQTL [19,71], which were defined as being composed of more than two original QTL, and having a physical distance < 1 Mb and a genetic map distance < 4 cM.

(3) As a third approach, we searched for candidate genes within the MQTL validated by GWAS-MTA, for those MQTL with more than three MTAs. For MQTL with long physical confidence interval (>1 Mb), a 1 Mb genomic region (500 kb on both sides of the MQTL peak) was used, for which the physical peak positions of the MQTL were calculated as proposed by Saini et al. [18]:(3)$$peak\;position\left(bp\right)=start\;position\left(bp\right)+\frac{\left(end\;position(bp)-start\;postion(bp)\right)}{\left(end\;position(cM)-start\;position(cM)\right)}\times \frac{CI(cM,95\%)}{2}$$
among them, peak position (bp): the center of physical positions of MQTL; start position (bp): left physical positions of MQTL; end position (bp): right physical positions of MQTL; start position (cM): left genetic position of MQTL; end position (bp): right genetic position of MQTL.

For the candidate genes obtained by the three methods, a functional analysis was performed using gene ontology (GO) enrichment on the GENE DENOVO cloud platform (https://www.omicshare.com/tools accessed on 1 January 2024) in order to understand the biological functions of the MQTL. Then, the Kyoto Encyclopedia of Genes and Genomes (KEGG) pathway enrichment analysis (https://www.omicshare.com/tools accessed on 1 January 2024) was used.

### 4.6. Analysis of Expression Patterns of Candidate Genes

In addition, an expression analysis was performed for selected candidate genes from the MQTL. The transcriptome data of multiple tissues of maize were downloaded from the website qTeller (https://qteller.maizegdb.org/ accessed on 1 January 2024). The whole transcriptome data cover 46 tissues/stages [102], including the embryo (16, 18, 20, 22, 24 DAP), endosperm (16, 18, 20, 22, 24 DAP), leaf (0, 12, 18, 24, 30 DAP-NOPOL), leaf (0, 12, 18, 24, 30 DAP-POL), whole seed (2, 6, 10, 14, 18, 22, 24 DAP), stem (V1, V2), anthers (R1), cob (R1, V18), tassel (V13, V18), whole primary root (7 d), whole root system (3, 7 d), tap root (Z1, Z2, Z3, Z4), brace root (V13), and crown root (V7, V13). The expression level of candidate genes was evaluated by transcript per million (TPM) value, and the genes with TPM value > 2 in tissue expression [103] were screened for expression analysis, and the $\log_{2}\left(TPM+1\right)$ was used for heat map plotting.

## 5. Conclusions

In this study, a total of 83 MQTL for drought tolerance in maize were identified and nearly half of them could be substantiated by results from genome-wide association studies. For the 20 core MQTL and maize–rice homologous genes, 583 candidate genes were identified. The MQTL and the candidate genes found in this study form the basis for future research on drought tolerance in maize and have the potential to assist the improvement in maize performance under drought conditions by molecular breeding.

## Figures and Tables

**Figure 1 ijms-25-04295-f001:**
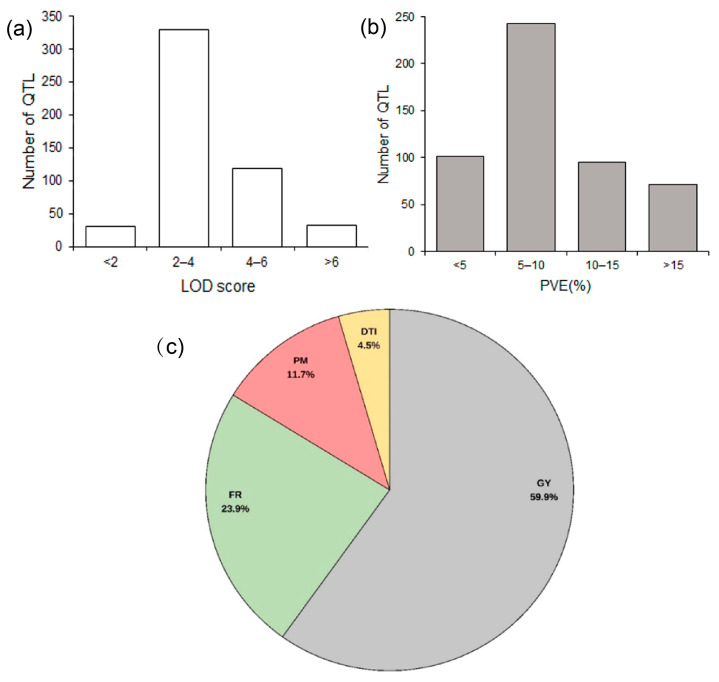
Classification of the QTL associated with drought used for this meta-analysis. (**a**) Distribution of their LOD score; (**b**) distribution of proportion of variance explained (PVE) by the QTL; (**c**) classification of the traits. GY, grain yield components traits; FR, flowering time-related traits; PM, plant morphology traits; DTI, drought tolerance index.

**Figure 2 ijms-25-04295-f002:**
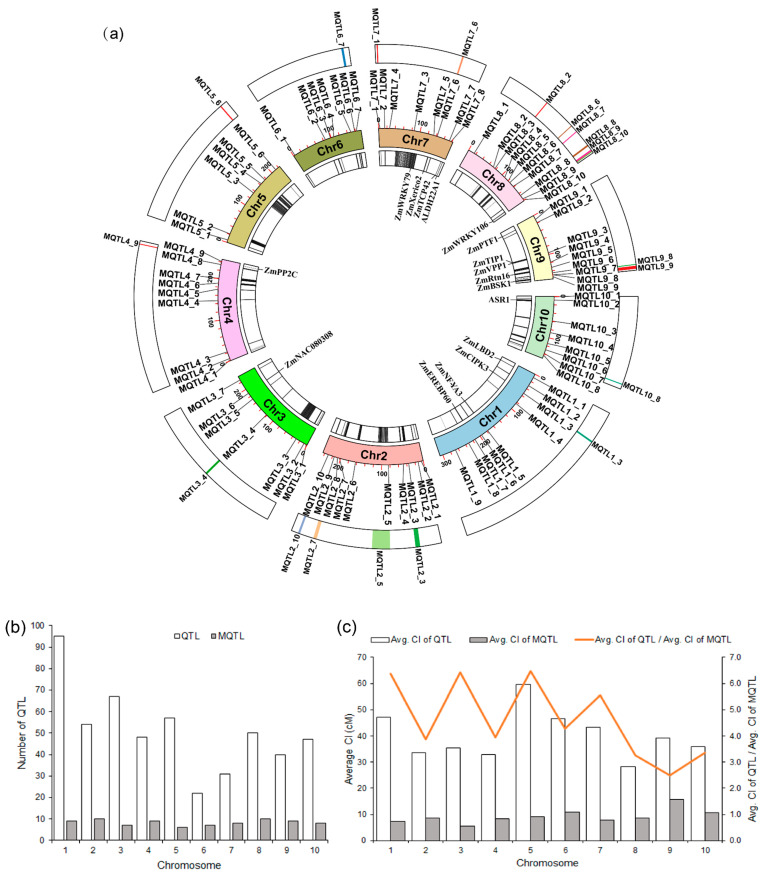
Characteristic features of QTL and MQTL. (**a**) Circular plot of the drought-related MQTL in maize. From the inside to the outside: the innermost circle represents the gene density contained in the MQTL as well as genes related to drought tolerance; the middle circle is the physical map position of the MQTL, and the outermost circle shows the core and hotspot MQTL intervals; (**b**) distribution of QTL and MQTL on the different maize chromosomes; (**c**) comparison of the confidence intervals of QTL and MQTL, showing the fold level of reduction in the size of the confidence interval.

**Figure 3 ijms-25-04295-f003:**
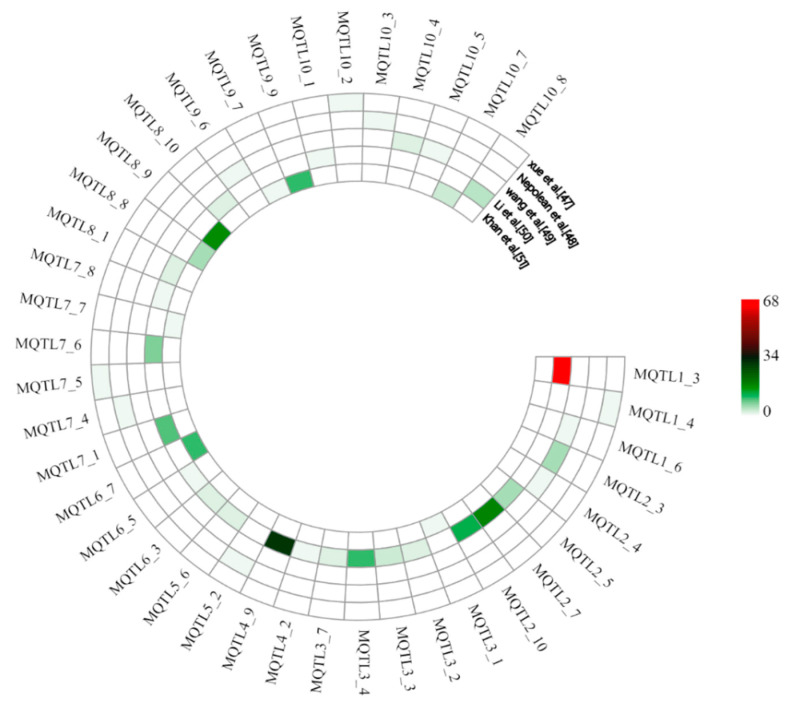
Support of identified MQTL by results from genome-wide association mapping studies with five different natural populations. Colors from red to green indicate the number of marker–trait associations overlapping with the MQTL.

**Figure 4 ijms-25-04295-f004:**
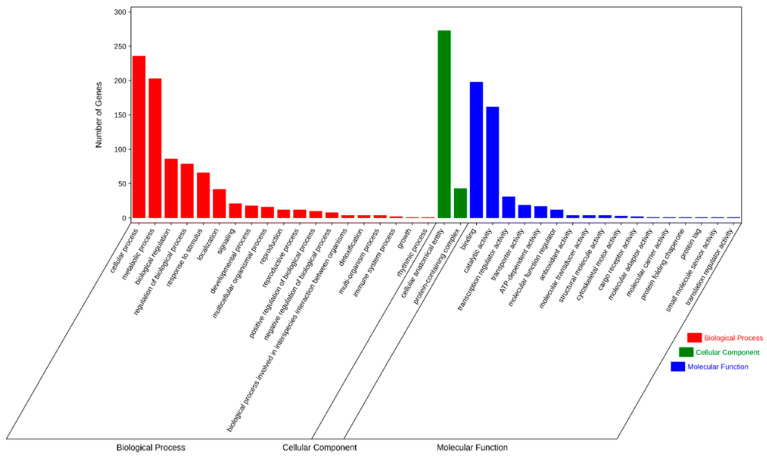
Gene ontology (GO) terms for the candidate genes identified in the MQTL regions.

**Figure 5 ijms-25-04295-f005:**
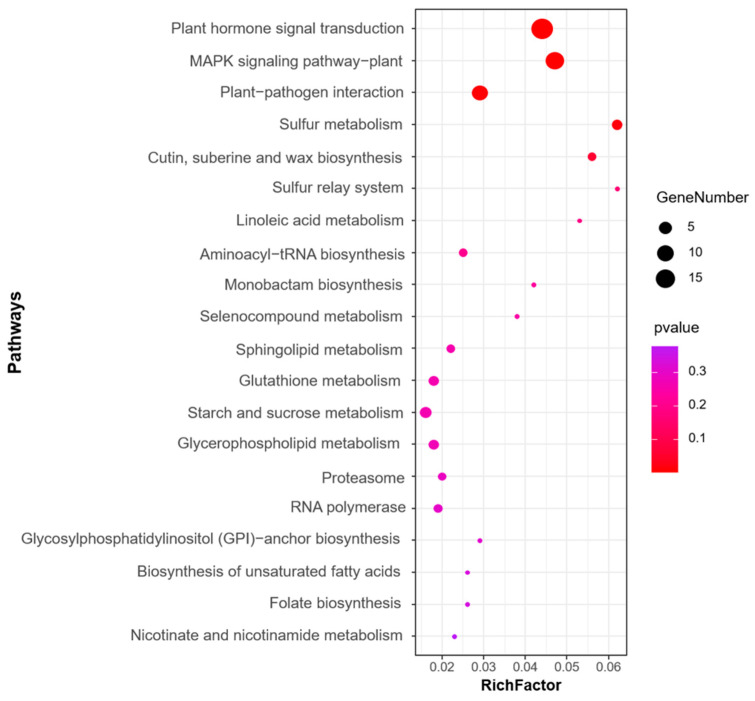
Top 20 KEGG enrichment pathways for the candidate genes identified for the MQTL regions.

**Figure 6 ijms-25-04295-f006:**
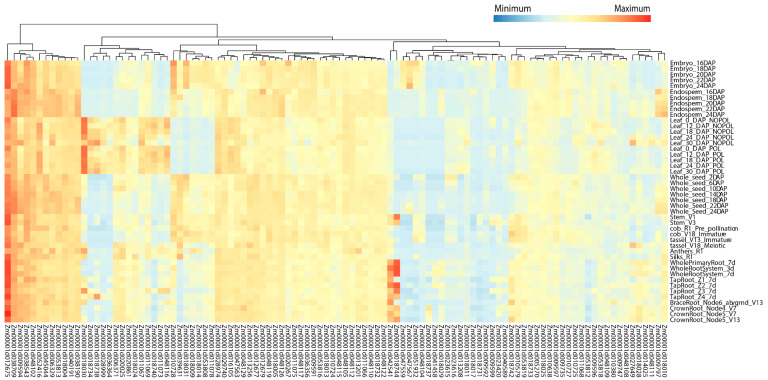
Heat map of high-confidence candidate genes expressed at ≥ 2 transcript per million. DAP, days after pollination; POL, pollination; NOPOL, no pollination; d, day; V, vegetative stages; R, reproductive stages; Z1, Zone 1 (first cm of root tip); Z2, Zone 2 (from end of Z1 to the point of root hair or lateral root initiation); Z3, Zone 3 (lower half of differentiation zone); Z4, Zone 4 (upper half of differentiation zone).

**Figure 7 ijms-25-04295-f007:**
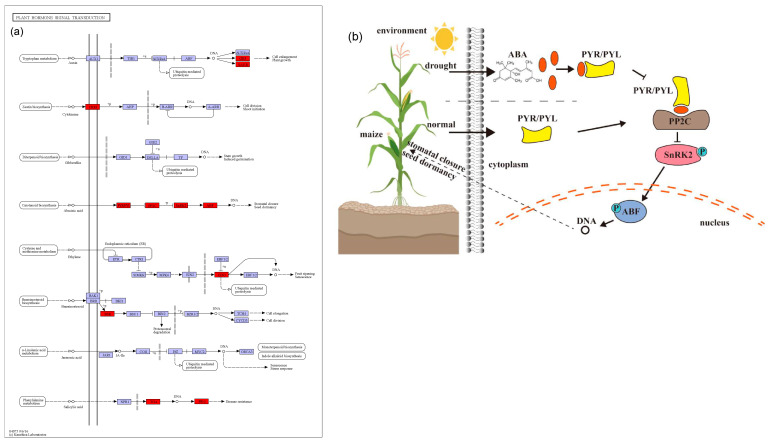
A model representing maize drought tolerance candidate genes associated with plant hormone signal transduction. (**a**) Plant hormone signal transduction (https://www.kegg.jp/pathway/ko04075 accessed on 1 October 2023). The red boxes represent the candidate genes involved in drought tolerance. (**b**) ABA signaling pathway.

**Table 1 ijms-25-04295-t001:** Candidate genes for the MQTL involved in plant hormone signal transduction pathways.

MQTL	Gene ID	Pathway	Description
MQTL1_3	*Zm00001d028974*	ETH	Ethylene insensitive-like3
MQTL2_10	*Zm00001d007395*	IAA	Auxin amido synthetase3
MQTL2_10	*Zm00001d007448*	SA	Pathogenesis-related protein19
MQTL4_9	*Zm00001d053815*	IAA	Small auxin up RNA45
MQTL5_1	*Zm00001d013201*	ABA	Serine/threonine-protein kinase SRK2E
MQTL5_3	*Zm00001d016105*	ABA	Abscisic acid receptor PYL10
MQTL5_3	*Zm00001d016294*	ABA	Abscisic acid receptor PYL3
MQTL7_1	*Zm00001d018734*	SA	Pathogenesis-related protein8
MQTL7_1	*Zm00001d018737*	SA	Pathogenesis-related protein13
MQTL7_1	*Zm00001d018738*	SA	Pathogenesis related protein4
MQTL7_4	*Zm00001d019364*	SA	Pathogenesis-related protein15
MQTL8_3	*Zm00001d009747*	ABA	Protein phosphatase homolog15
MQTL8_5	*Zm00001d010638*	ABA	bZIP-transcription factor 96
MQTL8_5	*Zm00001d010697*	IAA	Auxin amido synthetase12
MQTL8_8	*Zm00001d012005*	CTK	Putative histidine kinase family protein
MQTL8_9	*Zm00001d012538*	ABA	Abscisic acid-insensitive5-like protein 2
MQTL8_9	*Zm00001d012553*	SA	Octopine synthase binding factor4
MQTL9_5	*Zm00001d047563*	ETH	Ethylene insensitive-like1
MQTL9_9	*Zm00001d048345*	BRs	Brassinosteroid-signaling kinase1 bsk1

## Data Availability

All relevant data are within the paper and its Appendix A.

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
