# Peer review of "Meta-Quantitative Trait Loci Analysis and Candidate Gene Mining for Drought Tolerance-Associated Traits in Maize (Zea mays L.)"

_ijms, 2024, doi:10.3390/ijms25084295_

Round 1

Reviewer 1 Report

Comments and Suggestions for Authors

The manuscript presents a thorough examination of drought tolerance-associated traits in maize (Zea mays L.) using a meta-QTL analysis and candidate gene mining approach. The study aims to elucidate the genetic architecture of drought tolerance in maize by analyzing 511 quantitative trait loci (QTL) associated with various aspects of grain yield, flowering time, plant morphology under drought conditions, and a drought tolerance index from 27 previously published studies. The objectives are clearly stated, and the results are of interest to me. While I have no major comments, I have some suggestions outlined below:

Although the study provides valuable insights, the complexity of the genetic architecture and the large number of candidate genes identified may present challenges for direct application in breeding programs. Developing a simplified protocol to identify the major candidate genes would be beneficial. Further discussion on this matter would enhance the comprehensiveness of the study.

The study identifies candidate genes based on their proximity to MQTL and homology with known genes in other species. However, functional validation in maize is imperative to confirm their roles in drought tolerance. Therefore, the next step of this study should focus on this aspect. Additional discussion or outlining future plans would contribute to addressing this concern.

Relying solely on previously identified QTL and rice homologs might overlook novel genes and pathways specific to maize that contribute to drought tolerance. Incorporating information from other grass species could shed light on the differences between rice and maize.

Direct functional validation, consideration of novel mechanisms, and deeper exploration of gene interactions and environmental influences are necessary to fully understand the mechanisms of drought tolerance in maize.

Although the study identifies candidate genes and enriched pathways, it heavily relies on indirect evidence, such as gene orthology and presence within MQTL regions. Direct functional validation of these genes in maize under drought conditions is essential to confirm their roles in drought tolerance mechanisms.

The study does not extensively address how environmental variables or epigenetic modifications might influence the expression and function of identified candidate genes. Given the significant impact of environmental conditions on drought stress responses, further investigation in this area is crucial.

Comments on the Quality of English Language

The English writing in the manuscript is generally well-structured and clear, effectively communicating complex scientific concepts and findings. However, some sentences are overly complex or lengthy, making them difficult to follow. Breaking these into shorter, simpler sentences could improve readability. Certain phrases and concepts are repeated, which can be redundant. Streamlining these instances could make the writing more concise. Additionally, some sections could benefit from smoother transitions to guide the reader through the document's progression and the logical flow of arguments or findings.

Reviewer 2 Report

Comments and Suggestions for Authors

The article "Meta-QTL analysis and candidate gene mining for drought tolerance-associated traits in maize (Zea mays L.)" by authors Ronglan Li et al. aims to provide a basis for a better understanding of the genetic and molecular mechanisms of drought tolerance in maize as a crucial for future breeding and a more targeted exploitation of genetic resources. The authors have performed a meta-QTL analysis for drought tolerance in maize. They identified a total of 83 MQTLs for drought tolerance in maize. For the 20 most essential MQTLs, the scientific group identified 583 candidate genes. The MQTLs and the candidate genes found in the current study form a reasonable basis for future research on drought tolerance in maize. They can assist in improving maize performance under drought conditions through molecular breeding.

The article is written very well. It contains enough information in every part, making it easy to read and understand.

The quality of the Figures could be better!

Because the whole article's data are based on the results obtained from other scientists, such results in the Discussion section can be included in the Results section. For example, Table 1 and its explanation!

Reviewer 3 Report

Comments and Suggestions for Authors

Overall, this manuscript is well-prepared, with clear writing and figures that are easy to interpret.

My major concern is the choice of rice genome for homology search and comparison. The authors should introduce the rationale behind this selection or provide more justification in the discussion.

Sorghum is widely known as a closely related crop to maize, but with greater resistance to drought stress. I suggest the authors to include available sorghum resources, such as drought-related traits, mapping results, homologous gene functions in the literature to enrich the context of this manuscript.

Comments on the Quality of English Language

The manuscript is well written and the data are clearly presented.
